# Synthesis of extended polycyclic aromatic hydrocarbons by oxidative tandem spirocyclization and 1,2-aryl migration

Xuan Zhang[1], Zhanqiang Xu[1], Weili Si[1], Kazuaki Oniwa[1], Ming Bao[2], Yoshinori Yamamoto[1,2] & Tienan Jin[1,2]

The extended polycyclic aromatic hydrocarbons (PAHs) have received significant interdisciplinary attention due to their semiconducting applications in diverse organic electronics as well as intriguing structural interests of well-defined graphene segments. Herein, a highly efficient oxidative spirocyclization and 1,2-aryl migration tandem synthetic method for the construction of extended polyaromatic hydrocarbons (PAHs) has been developed. The CuCl-catalyst/PhCO$_3^t$Bu or DDQ oxidation system in the presence of trifluoroacetic acid enables the selective single-electron oxidation to take place preferentially at the more electron-rich alkene moiety of *o*-biphenylyl-substituted methylenefluorenes, giving rise to the subsequent tandem process. A variety of structurally diverse extended PAHs including functionalized dibenzo[*g,p*]chrysenes, benzo[*f*]naphtho[1,2-*s*]picene, hexabenzo[*a,c,fg,j,l,op*]tetracene, tetrabenzo[*a,c,f,m*]phenanthro[9,10-*k*]tetraphene, tetrabenzo[*a,c,f,k*]phenanthro[9,10-*m*]tetraphene, tetrabenzo[*a,c,f,o*]phenanthro[9,10-*m*]picene and *S*-type helicene have been readily synthesized.

[1] WPI-Advanced Institute for Materials Research (WPI-AIMR), Tohoku University, Sendai 980-8577, Japan. [2] State Key Laboratory of Fine Chemicals, Dalian University of Technology, Dalian 116012, China. Correspondence and requests for materials should be addressed to T.J. (email: tjin@m.tohoku.ac.jp).

The extended polycyclic aromatic hydrocarbons (PAHs) have received significant interdisciplinary attention due to their semiconducting applications in diverse organic electronics as well as intriguing structural interests of well-defined graphene segments[1–8]. A variety of synthetic methodologies for constructing various π-extended PAHs have been developed towards achieving high efficiency and novelty[5–15]. Among them, since the seminal advance has been made by Müllen and Spiess *et al.* towards synthesizing hexa-*peri*-hexabenzocoronene (*p*-HBC) through a highly efficient oxidative intramolecular cyclodehydrogenation (Scholl reaction) of hexaphenylbenzene[16,17], the selective oxidative aromatic coupling of unfunctionalized arenes with a net loss of hydrogens has become the most popular, straightforward and atom-economical synthetic strategy of various PAHs involving nanographene and graphene[5–9]. However, this methodology still has some problems such as unpredictable regioselectivities due to the different activity of aromatic C–H bonds, electronic property of arenes and stereo factors. In this context, the discovery of novel and expedient aromatization methods towards structurally diverse PAHs with selectivity control and high efficiency is highly desirable.

Recently, we have developed a novel FeCl₃-mediated oxidative dehydrogenative spirocyclization of 1,2-di(9H-fluoren-9-ylidene)-1,2-diphenylethane for the synthesis of a new class of the dispirolinked π-system DSFIIF (Fig. 1a)[18]. In this study, the conjugated diene moiety in 1,2-di(9H-fluoren-9-ylidene)-1,2-diphenylethane was predicted to be more electron-rich than the fluorenyl and phenyl moieties by the frontier molecular orbital calculation, which suggests that it would easily undergo twofold single-electron oxidation preferentially to form a dication species, followed by the intramolecular Friedel-Crafts reaction to afford the corresponding dispirocycle DSFIIF. In light of this observation, we designed a new olefin substrate 9-(biphenyl-2-ylmethylene)-9H-fluorene (BPMF), in which the alkene moiety was calculated to possess relatively higher electron density compared to other moieties (Fig. 1b and Supplementary Fig. 1). Inspired by this observation, we anticipated that if a radical cation forms selectively at the alkene moiety of BPMF by employing appropriate single-electron oxidation systems, the subsequent spirocyclization and 1,2-aryl migration tandem process may take place to give the desired twisted molecule dibenzo[*g,p*]chrysene (DBC), which is an intriguing holding block of discotic molecules, fluorescent and charge carrier-transporting materials. Diverse synthetic methods of the DBC derivatives have been reported, such as oxidation of electron-rich bis(biaryl)acetylenes, oxidation of electron-rich tetraarylethylenes, intramolecular aromatic C–H/C–Br coupling, double Suzuki-Miyaura coupling, and Pd-catalysed coupling of C–H bonds of small PAHs with dimethyldibenzosilole[13–15,19–28]. On the basis of our reaction design, we herein describe a novel dehydrogenative PAH synthetic methodology based on the oxidative spirocyclization and 1,2-aryl migration tandem process. A variety of extended PAHs involving functionalized DBCs, hexabenzotetracene, tetrabenzophenanthrotetraphene, tetrabenzophenanthropicene and helicene derivatives have been readily synthesized in good to high yields without prefunctionalization of aromatic rings.

**Figure 1 | Single-electron oxidation of alkenes for synthesis of π-conjugated polycyclic hydrocarbons. (a)** Our previous method for synthesis of dispirofluorene-indenoindenefluorene (DSFIIF) via FeCl₃-promoted oxidative spirocyclization of 1,2-di(9H-fluoren-9-ylidene)-1,2-diphenylethane (DFDPE). **(b)** This study for the single-electron oxidation induced spirocyclization and 1,2-aryl migration tandem synthesis of PAHs.

## Results

**Investigation of single-electron oxidation conditions.** Based on our previous oxidative spirocyclization[18], we firstly examined the weak acidic oxidant FeCl$_3$ using the readily available substrate **1a** (Table 1, see also Supplementary Methods). Unfortunately, the use of FeCl$_3$/FeO(OH) neutral systems resulted in no reaction, while the acidic FeCl$_3$ in $o$-xylene at 80 °C only produced the spirocyclic product **3a** (vide infra, Fig. 6) in 88% yield without formation of the desired product **2a**. After investigation of various copper catalysts (20 mol%) combined with molecular oxygen (1 atm) in the presence of trifluoroacetic acid (TFA; 5 equiv), we were pleased to find that the desired tandem reaction did proceed with CuCl catalyst/O$_2$ systems, affording the corresponding product **2a** in 39% yield (Table 1, entry 1). The use of other oxidants such as $tert$-butyl hydroperoxide (TBHP) and di-$tert$-butyl peroxide (DTBP) in place of O$_2$ have no beneficial effect in achieving a high yield of **2a** (entries 2 and 3). To our delight, the yield of **2a** was improved significantly when PhI(OAc)$_2$ and PhCO$_3{}^t$Bu were used as oxidants in conjunction with the CuCl catalyst (entries 4 and 5); particularly, the latter oxidant produced **2a** in nearly quantitative yield. The combination of a Cu(I) salt with PhCO$_3{}^t$Bu was reported to enable the formation of a PhCO$_2$Cu(II) salt along with a $^t$BuO radical species $in$ $situ$ to generate a allylic radical[29], thus suggesting the involvement of a radical mechanism in the present tandem reaction. The use of PhCO$_3{}^t$Bu, DTBP and PhI(OAc)$_2$ as oxidants in the absence of the copper catalyst led to poor yields of **2a** (entries 6–8), indicating the important role of the copper catalyst in the Cu(I)/oxidant systems. Further screening of other oxidants in the absence of the copper catalyst revealed that 2,3-dichloro-5,6-dicyanobenzoquinone (DDQ) and $o$-chloranil were also effective in achieving high yields of **2a** (entries 9 and 10). It was noted that the use of TFA as an additive was necessary for rendering this transformation; the strong acid such as trifluoromethanesulfonic acid (TfOH) gave the spirocycle **3a** in 95% yield, while the weak acid such as acetic acid resulted in no reaction. It has been demonstrated that the DDQ/strong acid system readily undergoes a single-electron oxidation of various electron donors with high oxidation potentials to afford the corresponding radical cation species[23,30–32], implying the present transformation involves the radical cation formation process.

**Synthesis of various PAHs with versatile functional groups.** With the optimized conditions obtained from entries 5 and 9 in Table 1 as conditions A and B, the electronic effect of substituents on the fluorene and biphenyl moieties in the starting BPMFs was investigated to understand the influence on the construction of the DBC scaffold (Fig. 2). The reaction of **1b** with two electron-rich $t$-butyl substituents on the fluorene moiety under condition A afforded a higher yield of the corresponding DBC **2b** than that under condition B. Likewise, the condition A showed higher activity than the condition B for the reactions of **1c** and **1d** with two electron-poor substituents such as Br and I as R$^1$. The electron-rich substrate **1b** showed much higher reactivity as compared to the electron-poor substrates **1c** and **1d** under both conditions A and B. Similar electronic effects were also observed from the substrates with substituents on the biphenyl moiety. For example, the reactions of **1e** and **1f** with two electron-rich substituents such as Me and MeO as R$^2$ produced the corresponding DBCs **2e** and **2f** in high yields under both conditions A and B, while the reactions of **1g** and **1h** having two electron-withdrawing substituents such as F and Cl as R$^2$ afforded the corresponding products **2g** and **2h** in lower yields with longer reaction times. The BPMF **1i** and its perdeuterated analogue **1i-$d_5$** bearing electron-donating $t$-butyl and methoxy groups at fluorene and biphenyl moieties showed a high reactivity, which underwent the tandem annulation at 40 °C under both conditions A and B to afford the corresponding DBC derivatives **2i** and **2i-$d_4$** in good yields and no D–H exchanged products were observed for the reaction of **1i-$d_5$**. In the reactions of the benzothiophene-substituted substrate **1j**, the DDQ-mediated condition B showed higher activity than the Cu-catalysed condition A, giving the corresponding DBC derivative **2j** in moderate yields. Interestingly, the tandem reaction of **1k** composed of binaphthyl instead of biphenyl also proceeded efficiently, giving rise to the helicene product **2k** in high yields[33].

The growing interests in the synthesis and potential application of extended PAHs as graphene molecules in optoelectronics led

---

**Table 1 | Optimization of oxidation conditions.**

| Entry | Cu catalyst | Oxidant | Yield of 2a (%)* |
|---|---|---|---|
| 1 | CuCl | O$_2$ (1 atm) | 39 |
| 2 | CuCl | TBHP | 7 |
| 3 | CuCl | DTBP | 16 |
| 4 | CuCl | PhI(OAc)$_2$ | 61 |
| 5† | CuCl | PhCO$_3{}^t$Bu | 100 (94) |
| 6 | – | PhCO$_3{}^t$Bu | 4 |
| 7 | – | DTBP | 2 |
| 8 | – | PhI(OAc)$_2$ | 45 |
| 9 | – | DDQ | 92 (87) |
| 10 | – | $o$-chloranil | 85 |

Reaction conditions: **1a** (0.2 mmol), CuCl (20 mol%), oxidant (1.5 equiv or 1 atm), TFA (5 equiv), $o$-xylene (1.2 ml), at 80 °C for 12 h.
*$^1$H NMR yield determined using dibromomethane as an internal standard. Isolated yields are shown in parentheses.
†The reaction time is 1.5 h.

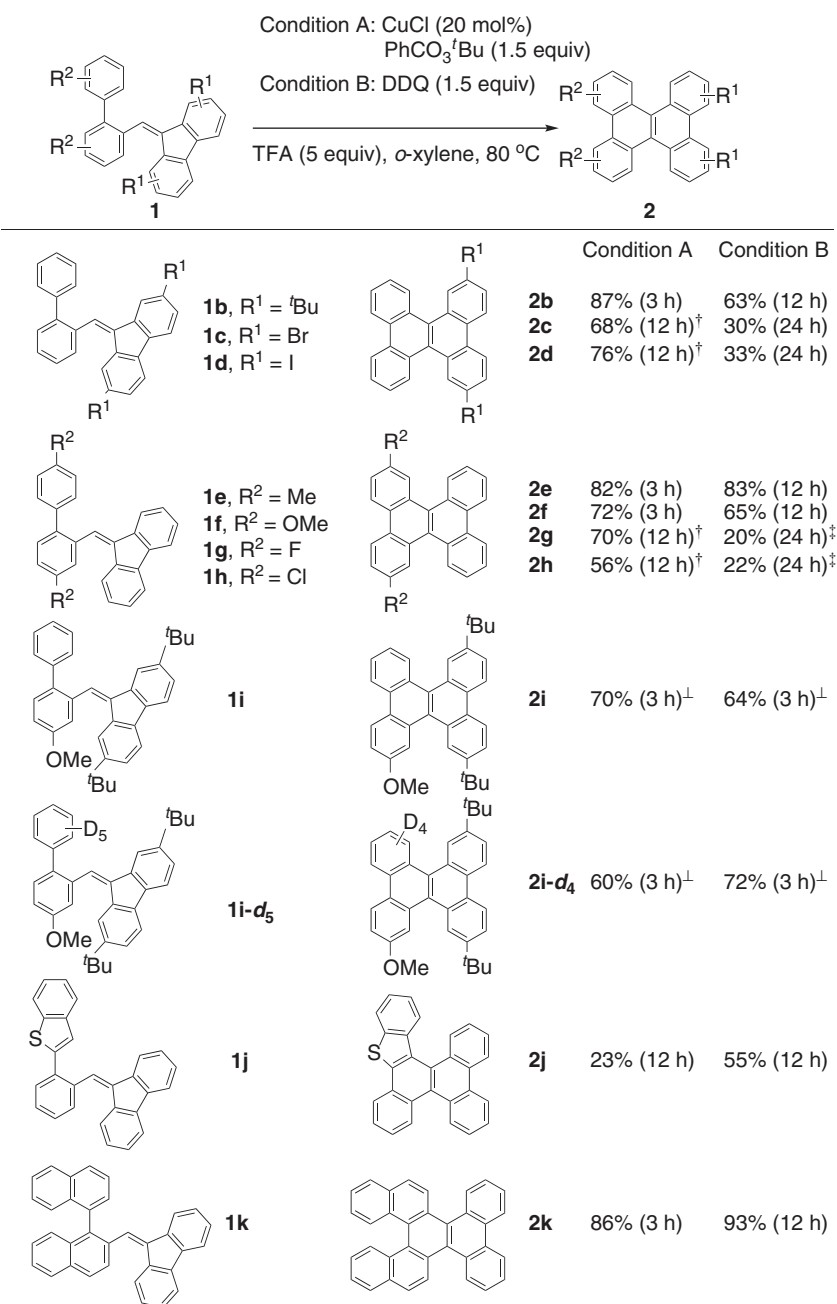

**Figure 2 | Substrate scope of oxidative aromatization.** The isolated yields are shown. †CuCl (40 mol%), PhCO₃ᵗBu (3 equiv) and TFA (10 equiv) were used. ‡DDQ (3 equiv) and TFA (10 equiv) were used. ⊥The reaction temperature is 40 °C.

us to examine the construction of highly extended PAHs using the present tandem reaction[1–4]. In contrast to the activity for the synthesis of the DBC scaffold, the CuCl-catalysed condition A was less active than the DDQ-mediated condition B for the construction of the following extended PAHs. Under condition A, the double annulation reaction of **1l** containing two biphenyl-centered methylenefluorene moieties produced the corresponding hexabenzo[*a,c,fg,j,l,op*]tetracene **2l** in a low yield of 10% (Fig. 3a). It was noted that the monoannulated product **2l′** was dominantly formed in 60% yield (see Supplementary Information for the structure of **2l′**). To our delight, the yield of **2l** increased to 55% under the DDQ-mediated condition B with the formation of **2l′** in 16% yield. The structure of **2l** was recently reported to possess an interesting helically twisted conformation[14]. We also

examined the double annulation of the *p*-terphenyl-centered methylenefluorene substrates **1m** and **1n** (Fig. 3b). The reaction of **1m** under condition B proceeded well to afford the corresponding tetrabenzo[*a,c,f,m*]phenanthro[9,10-*k*]tetraphene **2m** (ref. 34) in 88% yield, while only a 20% yield of **2m** was obtained under condition A. The combination of the CuCl catalyst with DDQ (condition C) further increased the yield of **2m** to 95%. Although the condition B was not sufficiently effective for the double annulation of the *t*-butyl-substituted substrate **1n**, fortunately, the condition C provided the corresponding product **2n** (ref. 13) in a high yield of 86%. The present method was also successfully applied to the synthesis of several novel π-extended PAHs. When the *m*-terphenyl-centered methylenefluorene **1o** was subjected to both conditions A and B,

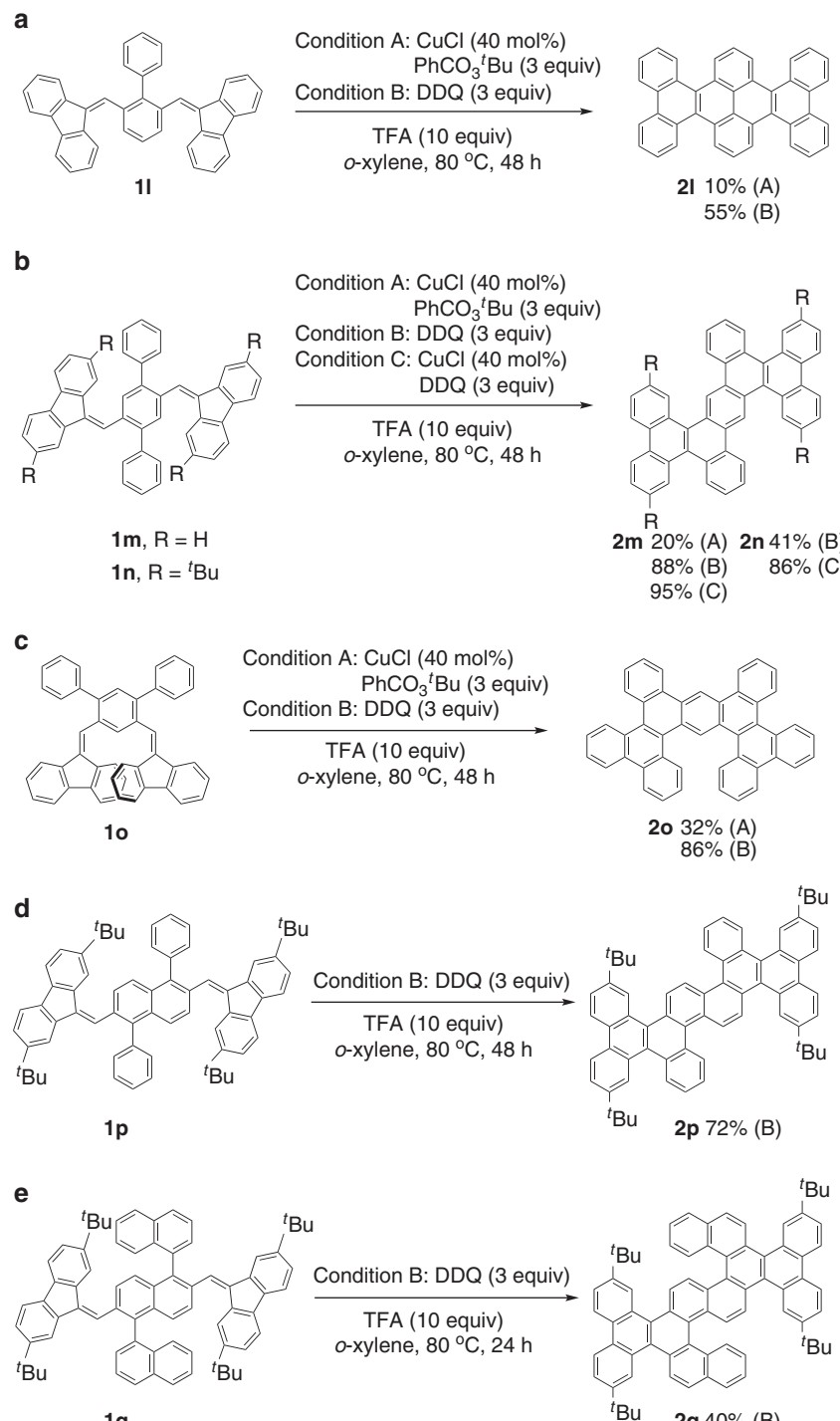

**Figure 3 | Synthesis of extended PAHs.** Double annulations of (**a**) biphenyl-centered bis-methylenefluorenes, (**b**) *p*-terphenyl-centered bis-methylenefluorenes, (**c**) *m*-terphenyl-centered bis-methylenefluorenes, (**d**) 1,5-diphenylnaphthalene-centered bis-methylenefluorenes and (**e**) 1,1':5',1''-ternaphthalene-centered bis-methylenefluorenes.

the corresponding new PAH of tetrabenzo[*a,c,f,k*]phenanthro [9,10-*m*]tetraphene **2o** was obtained in 32% and 86%, respectively (Fig. 3c). Moreover, the 1,5-diphenylnaphthalene-centered methylenefluorene **1p** also underwent the double tandem annulation efficiently under condition B to afford the novel π-extended PAH **2p** in 72% yield, while the condition A resulted in decomposition of **1p** (Fig. 3d). Remarkably, a novel π-extended *S*-type helicene product **2q** could be constructed in 40% yield upon the double annulation of the

1,1':5',1''-ternaphthalene-centered methylenefluorene **1q** under condition B (Fig. 3e). The different activity of conditions A to C awaits further clarification with respect to the yield and reactivity, but we assumed that the electron distribution of benzene components in various substrates should highly affect the single-electron oxidation of the alkene moiety.

We also examined the triaryl-substituted ethene **1r** bearing a diphenylmethylene unit instead of the fluorene moiety (Fig. 4). As expected, the reaction of **1r** underwent the present tandem

reaction efficiently, affording the corresponding 9,10-diphenyl-phenanthrene **2r** in high yields under both conditions A and B. Other biphenyl-substituted ethenes replacing the 2,2-diphenyl unit in **1r** with different substituents, such as Ph and Ph, Ph and Me, Ph and H, dimethyl and two hydrogens were examined to be totally inactive, demonstrating the importance of triaryl groups for the implementation of the present tandem transformation. The aryl groups are thought to be crucial for the radical cation stabilization and 1,2-aryl migration tandem process as shown in Fig. 7.

**Optical and electrochemical properties of 2k–q.** The ultraviolet–visible absorption of extended PAHs of **2p** (426 nm) and **2q** (432 nm) in diluted chloroform solution show red-shifted onsets compared to that of **2k–o** due to their large π-extension lengths (Table 2 and Fig. 5). The fluorescence spectra of the extended PAHs have emission maxima at the region of 436–475 nm and exhibit relatively large Stock shifts in the range of 17–65 nm, implying their weak structural rigidity. Interestingly, the *S*-type helicene **2q** exhibits a much smaller Stock shift of 17 nm compared to the simple helicene **2k** (65 nm), indicating the higher structural rigidity of the former. The compounds **2l**, **2m** and **2p** in diluted chloroform have moderate fluorescence quantum yields of 0.27, 0.36 and 0.31, respectively, which are higher than the compound **2o** (0.19). It was found that both helicene compounds **2k** (0.06) and **2q** (0.14) showed relatively low emission properties. The highest occupied molecular orbital (HOMO) energy level calculated from the oxidation potential by cyclic voltammetry (Supplementary Fig. 2) shows that the helicene **2k** has the lowest HOMO of $-5.65$ eV and **2o** has the highest HOMO of $-5.25$ eV compared to that of other four PAHs ($-5.42$ to $-5.50$ eV). The lowest unoccupied molecular orbital (LUMO) energy level estimated from the HOMO and the optical energy gap shows that **2k**, **2m**, **2p** and **2q** have lower LUMOs in the range of $-2.51$ to $-2.63$ eV compared to the PAHs **2l** ($-2.37$ eV) and **2o** ($-2.18$ eV). The large extended PAHs **2p** and **2q** show smaller HOMO–LUMO gaps than that of other PAHs, which is attributed to the decreased LUMO orbitals more than their HOMO orbitals.

## Discussion

To further understand the reaction pathways, the spirocyclic compound **3a** was used as a starting substrate under both conditions A and B (Fig. 6), which was prepared as a byproduct during the optimization of reaction conditions. Although the yields of the desired product **2a** were relatively low, the oxidation reactions did proceed under the standard conditions, suggesting the formation of spirocyclic radical **B** and cation **C** intermediates (Fig. 7) through the sequential single-electron oxidation of **3a**. The formation of the cation species **C** may give rise to subsequent 1,2-aryl shift to produce the final product **2a**. It should be mentioned that, however, the present tandem reaction of **1a** to **2a**

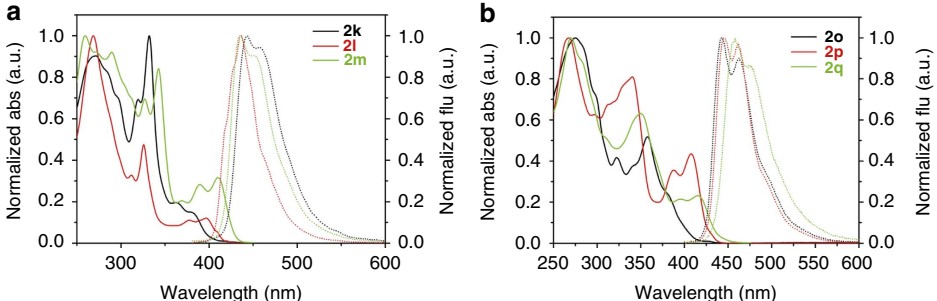

**Figure 4 | Construction of phenanthrene motif.** The reaction with nonfluorenyl substrate **1r** is able to produce the corresponding 9,10-diphenylphenanthrene **2r** under the standard conditions.

**Table 2 | Optical and electrochemical properties of 2k–q.**

| Compound | $\lambda_{onset}$ (nm)* | $\lambda_{em}^{max}$ ($\lambda_{exc}$) (nm)* | $\Phi_f$† | $\Delta E^{opt}$ (eV)‡ | HOMO (eV)§ | LUMO (eV)‖ |
|---|---|---|---|---|---|---|
| **2k** | 405 | 443, 456 (320) | 0.06 | 3.06 | $-5.65$ | $-2.59$ |
| **2l** | 406 | 437 (320) | 0.27 | 3.05 | $-5.42$ | $-2.37$ |
| **2m** | 418 | 436, 450 (340) | 0.36 | 2.97 | $-5.48$ | $-2.51$ |
| **2o** | 404 | 442, 462 (300) | 0.19 | 3.07 | $-5.25$ | $-2.18$ |
| **2p** | 426 | 444, 462 (300) | 0.31 | 2.91 | $-5.47$ | $-2.56$ |
| **2q** | 432 | 458, 475 (300) | 0.14 | 2.87 | $-5.50$ | $-2.63$ |

*Ultraviolet–visible absorption fluorescence spectra were measured in chloroform.
†Absolute fluorescence quantum yield ($\Phi_f$) was measured by a photon-counting method using an integration sphere.
‡Optical energy gap ($\Delta E^{opt}$) was estimated from the contact between the ultraviolet visible absorption and the fluorescence spectra according to the equation, $\Delta E^{opt}$ (eV) = 1,240/$\lambda_{onset}$.
§HOMO was calculated from the oxidation potential.
‖LUMO was calculated from the HOMO energy and the optical energy gap.

**Figure 5 | Ultraviolet–visible absorption and fluorescence spectra of extended PAHs.** The ultraviolet–visible absorption (solid line) and fluorescence spectra (dash line) of **2k–m** (**a**), and **2o–q** (**b**) in diluted chloroform. The ultraviolet–visible absorption red-shifted as expending their π-conjugation lengths. The extended PAHs possess relatively large Stock shifts, indicating their weak structural rigidity.

may not involve the formation of **3a** by the H atom abstraction. In addition, an intermolecular competing reactions between the protonated substrate **1i** and the perdeuterated substrate **1i-$d_5$** were studied in the same reaction vessel to understand the kinetic isotope effects (KIEs; Supplementary Fig. 3). When a 1:1 mixture of **1i** and **1i-$d_5$** was heated at 40 °C for 1 h under conditions A and B, respectively, the corresponding products **2i** and **2i-$d_4$** were obtained with similar KIE values of 1.21 and 1.20. The small isotope effect suggests that the present reaction involves a rapid aromatic deprotonation process and in other words, the C–C bond formation or cleavage steps can be presumed to be the rate-determining.

The proposed reaction mechanism is outlined in Figure 7 in terms of the experimental information, such as the indispensable role of single-electron oxidants, electronic effect of substrates, control reaction with the spirocyclic substrate and KIE values. Initially, a single-electron oxidation takes place at the more electron-rich alkene moiety of **1a** by the CuCl/PhCO$_3^t$Bu or DDQ oxidation system to form a radical cation species **A** (refs 9,35). Subsequently, the spirocyclization proceeds through the intramolecular Friedel-Craft reaction to afford a spirocyclic radical species **B**, which may undergo a second single-electron oxidation by the excess oxidants to form a spirocyclic cation **C**.

Next, the direct 1,2-aryl migration[36–38] of the cation **C** or more likely through an arenium cation intermediate **D** affords the cation **E**, which undergoes the rapid deprotonation to give the stable aromatic product **2a**.

In summary, we have disclosed a novel and efficient tandem synthetic method for the construction of various extended PAHs by the single-electron oxidation of various (o-biphenylyl)-methylene-substituted fluorenes. The selective single-electron oxidation of the alkene moiety by the Cu catalyst/PhCO$_3^t$Bu or DDQ oxidation system in the presence of TFA enabled the subsequent spirocyclization and 1,2-aryl migration tandem process to take place. Following this method, various known and new extended PAHs including the functionalized DBCs, benzo[f]naphtho[1,2-s]picene, hexabenzo[a,c,fg,j,l,op]tetracene, tetrabenzo[a,c,f,m]phenanthro[9,10-k]tetraphene, tetrabenzo[a,c,f,k]-phenanthro[9,10-m]tetraphene, tetrabenzo[a,c,f,o]phenanthro[9,10-m]picene and S-type helicene have been readily synthesized. Further extension of the present tandem synthetic strategy to a new class of π-extended polycycles with diverse ring systems is in progress.

## Methods

**Materials.** For NMR spectra of compounds in this manuscript, see Supplementary Figs 4–52. For the calculated electron distribution of HOMO orbital of **1a**, see Supplementary Fig. 1. For the cyclic voltammograms of compounds **2k–m** and **2o–q**, see supplementary Fig. 2. For the KIE experiments, see Supplementary Fig. 3. For the general information, experimental procedures and analytic data of compounds synthesized, see Supplementary Methods.

**General procedure for synthesis of 2a.** To a o-xylene (1.2 ml) solution of CuCl (4 mg, 20 mol %) and PhCO$_3^t$Bu (58.3 mg, 0.3 mmol) (condition A) or DDQ (68.1 mg, 0.3 mmol, condition B) were added TFA (114 mg, 1 mmol) and BPMF (**1a**) (66 mg, 0.2 mmol) at room temperature. The mixture was heated at 80 °C for 1.5 h (condition A) or 12 h (condition B). After cooling to room temperature, the reaction mixture was monitored by TLC and GC–MS. The reaction mixture was washed with water and extracted with CH$_2$Cl$_2$ for 2 times. After concentration of the CH$_2$Cl$_2$ solution, the resulting residue was purified by flash silica gel chromatography using a mixture of CH$_2$Cl$_2$/hexane as eluent to give the corresponding product **2a** in 94% yield (61.8 mg, condition A) or in 87% yield (57.2 mg, condition B) as a colourless solid.

Condition A: CuCl (20 mol%)
PhCO$_3^t$Bu (1.5 equiv)

Condition B: DDQ (1.5 equiv)

TFA (5 equiv), o-xylene, 80 °C, 12 h

**3a**

**2a** 9% (**3a**, 80%; A)
58% (**3a**, 42%; B)

**Figure 6 | Reactions using the spirocyclic substrate 3a for the formation of 2a.** Both single-electron oxidation conditions A and B afforded the corresponding product **2a**, indicating the involvement of spirocyclic radical species **B** and **C** for subsequent 1,2-aryl migration.

**Figure 7 | Proposed tandem reaction mechanism.** Initially, the reaction takes place via a single-electron oxidation of the relatively electron-rich alkene moiety of **1a**, followed by the intramolecular Friedel-Crafts reaction to form a spirocyclic radical species **B**, which is further oxidized to give a spirocyclic cation **C**. The cation **C** gives rise to the 1,2-aryl migration followed by aromatization to afford the desired dibenzochrysene **2a**.

**Data availability.** The authors declare that the data supporting the findings of this study are available within the paper and its Supplementary Information files, and also are available from the corresponding author upon reasonable request.

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

## Acknowledgements

This work was supported by JSPS KAKENHI grant number 15KK0180 on Fostering Joint International Research and World Premier International Research Center Initiative (WPI), MEXT, Japan.

## Author contributions

T.J. conceived the methodology and wrote the manuscript with the assistance of other authors. X.Z., Z.X. and W.S. conducted experiments. K.O. performed theoretical calculations. All the authors analysed the data.

## Additional information

**Competing interests:** The authors declare no competing financial interests.

**Publisher's note**: 

