## [Peer Review File · Nature Communications]

Reviewers' comments:

Reviewer #1 (Remarks to the Author):

This paper describes syntheses of dibenzo[g,p]chrysenes and other extended PAHs through a tandem reaction of oxidative spirocyclization and 1,2-aryl migration starting from 9-([1,1'-biphenyl]-2-ylmethylene)-9H-fluorene derivatives, which the authors have rationally designed based on their previous results. The authors have successfully optimized the reaction conditions to achieve the synthesis of dibenzo[g,p]chrysene in very high yields, and demonstrated the compatibility of the reaction with electron-donating and –withdrawing groups as well as the possibility of the double annulation to form tetrabenzo[a,c,f,m]phenanthro[9,10-k]tetraphene. However, although this reaction is new and interesting, there is no conclusive evidence for the reaction mechanism through radical cations which they propose. Moreover, the PAHs synthesized in this report are all previously reported and their properties are not outstanding. Therefore, I think this manuscript with the current state cannot meet the high standard of Nature Communications in addressing a general readership and a chemistry journal would be more appropriate. I nevertheless think it could still be possible to consider its publication in Nature Communications if the authors could properly address the following points:

1. The advantage of this method compared to the existing methods is unclear, especially because the current method requires relatively complicated substrates for making extended PAHs. Synthesis of novel PAHs by this method, which were not yet accomplished by the other methods, would improve the quality of this manuscript.
2. In analogy to the controversy regarding the reaction mechanism of the Scholl reaction, i.e., radical cation v.s. arenium cation, it is not possible to support the radical cation mechanism for the current reaction with the provided information. Indeed, the authors had to use a strong acid such as TFA and TfOH while acetic acid resulted in no reaction.
3. Additionally, I do not know if it is a problem of the file conversion, but the resolution of the NMR spectra in the supplementary information should be improved. It would be advisable to provide insets with magnified aromatic regions to show the purity of the compounds, especially because other purity proofs such as elemental analysis or HPLC data are not reported. This purity issue is a serious obstacle.

Reviewer #2 (Remarks to the Author):

The manuscript by Jin and coworkers is a nice report on a new method of preparing extended PAHs. The work uses oxidative condition to setup a aryl migration from a proposed spirocycle intermediate. The authors have provided many examples to demonstrate the scope of the reaction and have screened several oxidant conditions. I am welcoming of the manuscript as it provides a unique method to access these extended PAHs. My reservation is only in the impact of this work, especially for inclusion into Nature Communications. For example, many of the structures in this work are based on dibenzo[g,p]chrysenes. I would say there are recent examples that access similar types of materials, maybe even with starting materials that are more easily accessible than this work (e.g., Synlett 2015, 26, 1991–1996). The supporting information seems reasonable to repeat these experiments. I think the work is nice, just not as impactful as I don't see others using similar building blocks to apply such chemistry.

Reviewer #3 (Remarks to the Author):

Referee's report on the manuscript entitled:

Oxidative spirocyclization and 1,2-aryl migration tandem synthesis of extended polycyclic aromatic hydrocarbons

by Xuan Zhang, Zhanqiang Xu, Weili Si, Kazuaki Oniwa, Ming Bao, Yoshinori Yamamoto and Tienan Jin*

submitted to *Nature Communications* [NCOMMS-16-23511]

I consider this manuscript very important and very well written. The authors report on a novel method for the directed construction of extended polycyclic aromatic hydrocarbons by one-pot/two-step oxidative cyclisation synthesis. The method is based on readily accessible 9-(biphenyl-2-yl)methylenefluorenes which undergo a well-controllable sequence of spirocyclisation, induced by single-electron transfer, followed by 1,2-aryl shift. The method has been optimised and proved to be highly efficient, high-yielding, and broadly applicable. I am sure that it will contribute notably to the methodological arsenal for synthesising not only large polycondensed aromatic carbon networks, as the authors point out, but also other extended polyaromatic structures that contain also non-hexagonal rings. Therefore, I believe that the authors' results will gain a large degree of attention by scientists in organic chemistry and material sciences.

To provide an incisive assessment beyond that given above, I give concrete answers to some of the questions raised by the Editors:

Are they [the claims] novel and will they be of interest to others in the community and the wider field? – Yes, there are, to the best of my knowledge, and they are complementary to other methods.

Is the work convincing? – Yes, it is. The origin or, respectively, the synthesis access to starting 9-(biphenyl-2-yl)methylenefluorene derivatives are well documented and each of their chemical conversions are well described. The new compounds have been identified properly by NMR spectroscopic and accurate mass measurements performed by high-resolution mass spectrometry. Overall, the paper is very well written and presented.

On a more subjective note, do you feel that the paper will influence thinking in the field? – I can imagine that the paper will inspire many readers to think in a wider view on the possibilities to construct polycyclic aromatic and polyaromatic networks. Amusingly, the authors use building blocks that bear five-membered rings, i.e., fluorine derivatives, to elegantly convert them to six-membered rings within a greater polyhexagonal scaffold; this is in remarkable contrast to the topical construction on polycyclic aromatic hydrocarbons that contain nonhexagonal rings. Therefore, the paper represents an example of wider thinking. – Given that applications of the new method in many directions are conceivable, I am optimistic in the future objective *and* subjective influence of the paper.

We would also be grateful if you could comment on the appropriateness and validity of any statistical analysis, as well the ability of a researcher to reproduce the work, given the level of detail provided. – I have no doubts that the work is reproducible by skilled researchers.

Specific points of critics:

Main text

Finally, I have a quite a number of (mostly minor) suggestions for improvement and/or correction of the manuscript:

(1) Page 1, title: I find the title somewhat odd. I feel it could be improved, maybe by writing "**Synthesis of extended polycyclic aromatic hydrocarbons by oxidative tandem spirocyclization and 1,2-aryl migration**".

(2) Page 2, line 26, and elsewhere: I think that nomenclature has to be corrected in some instances (see also below). Here, the term "*o*-biphenyl-substituted methylenefluorenes" should rather become "*o*-biphenyl-yl-substituted methylenefluorenes".

(3) Page 3, lines 48-52: The sentence "In this study, the conjugated diene moiety in DFDPE was predicted to be more electron rich than the fluorenyl and phenyl moieties by the frontier molecular orbital calculation, which undergoes the double single electron oxidation preferentially to form a dication species, followed by the intramolecular Friedel-Craft reaction to afford the corresponding dispirocycle DSFIIF." contains several minor mistakes and has also wrong syntax. Rather, it should read: "In this study, the conjugated diene moiety in DFDPE was predicted to be more electron-rich than the fluorenyl and phenyl moieties by the frontier molecular orbital calculation, which suggests that it would easily undergo two-fold single-electron oxidation preferentially to form a dication species, followed by the intramolecular Friedel-Crafts reaction to afford the corresponding dispirocycle DSFIIF."

(4) Page 3, line 59: The term "carrier-transporting materials" is incomplete, I assume that the authors mean "charge carrier-transporting materials".

(5) Page 3, line 68: Also, I consider the phrase "prefunctionalisation of aromatic C-H bonds" inappropriate. It is not the "C-H bonds" but the "aromatic rings", "aromatic units", or aromatic nuclei" can be functionalised.

(6) Page 4, Figure 1: Although a complete mechanistic interpretation will follow at the end of the manuscript (Figure 7), I find the schematic presentation of the reaction sequence not clear enough. (i) The label "1,2-aryl migration" in the third step is not sufficient; it should read "single electron oxidation and 1,2-aryl migration". (ii) Also, the three steps ("single electron oxidation", spirocycliz(s)ation" and "single electron oxidation and 1,2-aryl migration") should be complemented by adding " $- e^-$ ", " $- H^+$ ", and " $- e^-$, $- H^+$ " near to the reaction arrows.

(7) Page 4, line 79: After "substrate **1a**", it should read "(Table 1, see also Supplementary Information)", instead of simply "(see Supplementary Information)". The latter version would irritate the reader who immediately wants to know the identity of "substrate **1a**".

(8) Page 5, line 88: "latter" (typo).

- (9) Page 5, lines 96-97: "TFA" and "TfOH" should be named explicitly; thus, "trifluoroacetic acid (TFA)" and "trifluoromethylsulfonic acid (TfOH)" or, maybe, "triflic (TfOH)". (It appears that "TfOH" does not appear anywhere later in the manuscript.)
- (10) Page 5, line 108: I think that "has been investigated" should rather be "was investigated".
- (11) Page 6: There are several minor mistakes. (i) "Br and I at R¹" should be "Br and I as R¹"; (ii) "Similar electronic effects were ..." would be better (line 114); (iii) "Me and OMe at R²" should be "Me and OMe as R²"; (iv) "F and Cl at R²" should be "F and Cl as R²"; (v) "The BPMF **1i** with a perdeuterated *o*-phenyl group" would be better; (vi) benzothiophene-substituted" (hyphen).
- (12) Page 7, line 131: Insert "the" before "synthesis".
- (13) Page 8, lines 135-136: "... composed of biphenyl-centered two methylenefluorene moieties ..." is poor English; I suggest "... containing two biphenyl-centered methylenefluorene moieties ...".
- (14) Page 8, lines 137-138: The sentence "It was noted ..." is too long; I suggest to delete the phrase "other than the corresponding product **1i**".
- (15) Page 8, legend to Figure 3: "... biphenyl-centered two methylenefluorenes ..." is again poor English; I suggest to use rational chemical nomenclature, for example, "... biphenyl-centered bis-methylenefluorenes ...". The same holds true for part (b) of the legend.
- (16) Page 9, line 157: I do not understand the term "trisubstituted 2-vinyl-1,1'-biphenyl" for compound **1o**. Why not "triaryl-substituted ethene"? – Also, the term "2,2-diphenyl group" does not make sense. Maybe "diphenylmethylene unit" would be appropriate.
- (17) Page 9, line 160-161: The sentence "It was noted that other 2-vinyl-1,1'-biphenyls by replacement of 2,2-diphenyl in **1o** with other substituents, such as Ph and Me, Ph and H, dimethyl, and two hydrogens were examined to be totally inactive, demonstrating the importance of aryl groups for the implementation of the present tandem transformation." is again quite poor and should be improved.
- (18) Importantly (see also below), I consider the name "1,1'-biphenyl" for the biphenyl unit unnecessarily complicated. The biphenyl unit is necessarily constructed by a C-C bond between two carbons of each ring. Therefore, I recommend avoiding the term "1,1'-biphenyl" and replace it by "biphenyl" in the whole manuscript, including the Experiment section in the Supplementary Information.
- (19) Page 9, line 163: "Maybe "crucial for" ("the radical ion stabilization") would be better than "related to".
- (20) Page 9, line 170: Delete "the" before "diluted".
- (21) Page 10, "Mechanistic investigation" (I): I do not fully agree with the discussion concerning the role of the spirocyclic intermediate(s). – Of course, the finding that the spirocyclic hydrocarbon **3a** can be converted into the dibenzocrysene **2a** supports – I would say, indirectly – the intermediacy of the related spirocyclic radical **B** shown in Figure 7. However, I think that the latter radical, being formed under the reaction conditions of the conversion **1a** → **2a**, does not undergo saturation (by H atom abstraction) to give **3a**; rather, **3a** has to be oxidised and converted to either cation **C** (Figure 7) or, at least, to the radical cation **3a^{•+}** in order to undergo the 1,2-aryl shift. Therefore, the

statement (in lines 198-201) "Although the yields of the desired product **2a** were relatively low, the oxidation reactions did proceed under the standard conditions, suggesting the formation of a spirocyclic radical intermediate in the present tandem reaction." is a bit too simple and should be modified or completed according to my above arguments.

(22) Page 10, "Mechanistic investigation" (II): The information gained from the kinetic isotope effects is not discussed clearly. – (i) The KIE values of 0.9 and 0.98 are similar and close to unity, but they are also smaller than unity. What is the error limit? – (ii) Do those values mean that there is, actually, no kinetic isotope effect? In fact, this can be presumed for the mechanism suggested in Figure 7, with the C-C bond formation (or cleavage) steps are rate-determining. – (iii) However, if the KIEs are reliably smaller than unity, this would be a counter-intuitive result, which should be commented. – (iv) From the Experimental section in the Supplementary Information, it appears that the KIN values were determined from separated runs with **1a** and **1i**. In that case, I would assume rather substantial experimental uncertainties. However, if the reactions of **1a** and **1i** were done in the very same solution, then the reliability should be quite high. In any case, the experimental setup should be also stated more clearly.

(22) Page 12, lines 224-225: The sentence "In summary, we have disclosed a novel and efficient tandem synthetic method of various extended PAHs by the single electron oxidation of alkenes tethered with *o*-biphenyl and fluorene moieties." is not clear and precise. – I would suggest writing "In summary, we have disclosed a novel and efficient tandem synthetic method for the construction of various extended PAHs by the single electron oxidation of various 1-*o*-biphenyl-2-fluorene-9-ylethenes." or, alternatively, "In summary, we have disclosed a novel and efficient tandem synthetic method for the construction of various extended PAHs by the single electron oxidation of various (*o*-biphenyl)methylene-substituted fluorenes."

(23) Page 12, line 243, and throughout the Experimental section (SI): The term "white solid" should be replaced by "colourless solid".

Supplemental Information

(24) Page S2, 1st paragraph: Instead of "High-resolution mass spectra were obtained on a BRUKER APEXIII spectrometer and JEOL JMS-700 MStation operator.", I recommend writing "Accurate mass data were obtained by high-resolution mass spectrometry performed on a BRUKER APEXIII spectrometer and JEOL JMS-700 MStation operator." (why the word "operator"?). In fact, the data obtained are accurate mass data; high-resolution is not an indispensable condition for obtaining accurate mass information; however, use of high resolution mass spectrometry is helpful if isobaric ions overlap.

(25) Page S2, 1st paragraph: The MALDI matrix used should be given also.

(26) Page S2, end of 2nd paragraph: The sentence "Other starting substrates and products were determined by ¹H, ¹³C NMR, and high-resolution mass." has poor style. It should read "Other starting substrates and products were determined by ¹H and ¹³C NMR spectroscopy and high-resolution mass spectrometry."

(27) Page S2, heading of 4th paragraph: Again, the naming of the biphenyl unit in the compound names should simply be "biphenyl" rather than "1,1'-biphenyl" – throughout the whole SI.

(28) Page S2, line 1 of text: To be strict, I wonder whether cupric chloride (CuCl) is soluble in *o*-xylene. However, I assume that it is soluble in a mixture of *o*-xylene and TFA.

(29) Page S2, line 7 of text: Once again, please note here and below the "white solid" should be changed into "colourless solid".

(30) Page S3, scheme to 2nd paragraph: I recommend changing the poor style of the aldehyde group (" =O ") into the better one (" -CHO "), as it appears in the next paragraph, for example.

(31) Page S5: From the presentation of the experiments, it appears indeed that two sets of two separate runs were performed. If this is really true, then the experimental uncertainties of the KIE values obtained should be rather substantial.

(32) Page S7, and throughout: To be strict again, I recommend making clear whether the calculated accurate mass values rely to the cationic species, e.g. $C_{26}H_{18}^+$ or, rather, $C_{26}H_{18}^{+\bullet}$ in the present case, or to the neutral species (which would be less appropriate).

(33) Page S7, and throughout: The experimentally obtained accurate mass data appear to be excellent throughout. In such cases, I would prefer to see the next decimal (fifth) to have information about the error limit. This is only a recommendation.

This is a long list of to-be-improved items, which is given to improve what is going to be an excellent paper. I think the paper does deserve this effort.

Reviewer: 1

Comments: This paper describes syntheses of dibenzo[g,p]chrysenes and other extended PAHs through a tandem reaction of oxidative spirocyclization and 1,2-aryl migration starting from 9-([1,1'-biphenyl]-2-ylmethylene)-9H-fluorene derivatives, which the authors have rationally designed based on their previous results. The authors have successfully optimized the reaction conditions to achieve the synthesis of dibenzo[g,p]chrysene in very high yields, and demonstrated the compatibility of the reaction with electron-donating and –withdrawing groups as well as the possibility of the double annulation to form tetrabenzo[a,c,f,m]phenanthro[9,10-k]tetraphene. However, although this reaction is new and interesting, there is no conclusive evidence for the reaction mechanism through radical cations which they propose. Moreover, the PAHs synthesized in this report are all previously reported and their properties are not outstanding. Therefore, I think this manuscript with the current state cannot meet the high standard of Nature Communications in addressing a general readership and a chemistry journal would be more appropriate. I nevertheless think it could still be possible to consider its publication in Nature Communications if the authors could properly address the following points:

Comment 1. The advantage of this method compared to the existing methods is unclear, especially because the current method requires relatively complicated substrates for making extended PAHs. Synthesis of novel PAHs by this method, which were not yet accomplished by the other methods, would improve the quality of this manuscript.

Response: Although the starting substrates used in this reaction seem to be complicated, these substrates can be prepared by a simple condensation of aldehyde and fluorene. The use of this method could construct various known and unknown extended PAHs. To prove the advantage of the present method on the construction of novel PAHs that cannot be prepared by the existing methods, we successfully synthesized three novel extended PAHs **2o**, **2p**, and **2q** in the supplementary experiments, which are shown in Figures 3c~3e and their elucidation are shown in page 8 in the revised manuscript. Moreover, their optical and electrochemical properties are discussed combined with the previous compounds **2k**, **2l**, and **2m** in page 10 in the revised manuscript. The data are summarized in Table 2 and UV-vis together with the fluorescence spectra are shown as Figure 5b. The synthetic

procedures of their precursors, analytic data of both starting substrates and products, NMR spectra, and CV charts (Figure S3) were provided in the revised Supplementary Information.

We very much appreciate this pertinent comment because the additionally synthesized novel PAHs would apparently improve the quality of our manuscript. Moreover, during this supplementary experiment, we were aware of the possibility of the present method in construction of elegant and novel PAHs, which will be studied at next stage and reported in the near future.

Comment 2. In analogy to the controversy regarding the reaction mechanism of the Scholl reaction, i.e., radical cation v.s. arenium cation, it is not possible to support the radical cation mechanism for the current reaction with the provided information. Indeed, the authors had to use a strong acid such as TFA and TfOH while acetic acid resulted in no reaction.

Response: Yes, the discussion regarding to the radical cation and arenium cation in the Scholl reaction is a continuous controversy. Although we couldn't provide the direct evidence for the radical cation species formation in the present reaction due to the difficulty of the isolation and detection of the radical species, we proposed the radical cation formation mechanism on the basis of the following indirect information.

(a) It has been demonstrated that the combination of DDQ with strong acids such as trifluoroacetic acid or methanesulfonic acid is able to oxidize various electron donors with the oxidation potentials as high as 1.7 eV via the single electron oxidation to form radical cation species. It is well agreement with our results obtained in the present reaction. Thus, we newly added the following sentence in page 5 in the revised manuscript by citing several new related literatures as refs. 33 to 35; "It has been demonstrated that the DDQ/strong acid system readily undergoes a single electron oxidation of various electron donors with high oxidation potentials to afford the corresponding radical cation species,^{26,33-35} implying the present transformation involves the radical cation formation process."

(b) The single electron oxidation usually proceeds preferentially with the electron donor having the relatively high overall electron density and occurs preferentially at the position of electron donors with the highest electron density. As we mentioned in the Introduction, the DFT calculation of BPFM indicates (also see Figure S1 in Supplementary information) that the alkene moiety in the starting molecule BPFM has the relatively higher electron density than other moieties. Thus we propose that the radical cation species may form preferentially at the alkene moiety of BPFM.

(c) Our previous work on spirocyclization of DFDPE having a methylenefluorene moiety elucidated in the Introduction part also indicates that the single electron oxidation takes place selectively at the alkene moiety of the methylene fluorene.

(d) It was reported that the electron-rich alkene such as 4,4'-(2-phenylethene-1,1-diyl)bis(N,N-dimethylaniline) readily underwent the single electron oxidation at the alkene position with I₂ oxidant to produce the corresponding radical cation species, which undergoes subsequent radical dimerization to form a stable and isolable dication intermediate. This literature was cited as a new ref. 38 at the mechanism part in page 12 in the revised manuscript.

Comment 3. Additionally, I do not know if it is a problem of the file conversion, but the resolution of the NMR spectra in the supplementary information should be improved. It would be advisable to provide insets with magnified aromatic regions to show the purity of the compounds, especially because other purity proofs such as elemental analysis or HPLC data are not reported. This purity issue is a serious obstacle.

Response: According to this comment, we provided insets with magnified ^1H and ^{13}C NMR spectroscopy for every compound in the revised Supplementary Information. We believe that all prepared compounds have enough high purity for publication in this journal.

Reviewer: 2

Comments: The manuscript by Jin and coworkers is a nice report on a new method of preparing extended PAHs. The work uses oxidative condition to setup a aryl migration from a proposed spirocycle intermediate. The authors have provided many examples to demonstrate the scope of the reaction and have screened several oxidant conditions. I am welcoming of the manuscript as it provides a unique method to access these extended PAHs. My reservation is only in the impact of this work, especially for inclusion into Nature Communications. For example, many of the structures in this work are based on dibenzo[g,p]chrysenes. I would say there are recent examples that access similar types of materials, maybe even with starting materials that are more easily accessible than this work (e.g., Synlett 2015, 26, 1991–1996). The supporting information seems reasonable to repeat these experiments. I think the work is nice, just not as impactful as I don't see others using similar building blocks to apply such chemistry.

Responses: The synthetic methods of our starting materials are very simple, which can be readily synthesized in high chemical yields in few synthetic steps. The literature commented by this reviewer mainly focused on synthesis of functionalized phenanthrenes by using a modified Pd-catalyzed annulation of 2,2'-bromobiphenyls with alkynes, in which two dibenzo[g,p]chrysenes were synthesized through the conversion of the corresponding 9,19-diaryl-substituted phenanthrenes under the typical Scholl reaction conditions [CuCl_2 (5 equiv), AlCl_3 (5 equiv)]. This literature was cited as a new ref. 31 in page 3.

Our work presented in this paper investigated a novel synthetic method aiming to the construction of various extended PAHs based on the optimization of the reaction conditions for constructing dibenzo[g,p]chrysene. Our paper introduced an alternative and novel synthetic way to construct various known and unknown PAHs by conversion of five-membered rings to six-membered rings, which may provide many readers a wide thinking on the possibilities for constructing a variety of new extended polyaromatic networks. We believe that our new observation may give rise to a large degree of attention for the researchers in synthetic chemistry and materials science.

Reviewer: 3

Comment: I consider this manuscript very important and very well written. The authors report on a novel method for the directed construction of extended polycyclic aromatic hydrocarbons by one-pot/two-step

oxidative cyclisation synthesis. The method is based on readily accessible 9-(biphenyl-2-yl)methylenefluorenes which undergo a well-controllable sequence of spirocyclisation, induced by single electron transfer, followed by 1,2-aryl shift. The method has been optimised and proved to be highly efficient, high-yielding, and broadly applicable. I am sure that it will contribute notably to the methodological arsenal for synthesising not only large polycondensed aromatic carbon networks, as the authors point out, but also other extended polyaromatic structures that contain also non-hexagonal rings. Therefore, I believe that the authors' results will gain a large degree of attention by scientists in organic chemistry and material sciences.

To provide an incisive assessment beyond that given above, I give concrete answers to some of the questions raised by the Editors:

Are they [the claims] novel and will they be of interest to others in the community and the wider field? – Yes, there are, to the best of my knowledge, and they are complementary to other methods.

Is the work convincing? – Yes, it is. The origin or, respectively, the synthesis access to starting 9-(biphenyl-2-yl)methylenefluorene derivatives are well documented and each of their chemical conversions are well described. The new compounds have been identified properly by NMR spectroscopic and accurate mass measurements performed by high-resolution mass spectrometry. Overall, the paper is very well written and presented.

On a more subjective note, do you feel that the paper will influence thinking in the field? – I can imagine that the paper will inspire many readers to think in a wider view on the possibilities to construct polycyclic aromatic and polyaromatic networks. Amusingly, the authors use building blocks that bear five-membered rings, i.e., fluorine derivatives, to elegantly convert them to six-membered rings within a greater polyhexagonal scaffold; this is in remarkable contrast to the topical construction on polycyclic aromatic hydrocarbons that contain nonhexagonal rings. Therefore, the paper represents an example of wider thinking. – Given that applications of the new method in many directions are conceivable, I am optimistic in the future objective and subjective influence of the paper.

We would also be grateful if you could comment on the appropriateness and validity of any statistical analysis, as well the ability of a researcher to reproduce the work, given the level of detail provided. – I have no doubts that the work is reproducible by skilled researchers.

Response: We very much appreciate for giving this high evaluation to our paper. It encourages us to continue this research area deeply and we believe that further elegant PAHs will be synthesized and reported in the near future.

Comments:

Main text

Finally, I have a quite a number of (mostly minor) suggestions for improvement and/or correction of the manuscript:

Comment (1) Page 1, title: I find the title somewhat odd. I feel it could be improved, maybe by writing “**Synthesis of extended polycyclic aromatic hydrocarbons by oxidative tandem spirocyclization and**

1,2-aryl migration”.

Response: Thank you for this pertinent comment and we agree to change the title as commented; “**Synthesis of extended polycyclic aromatic hydrocarbons by oxidative tandem spirocyclization and 1,2-aryl migration**”.

The changes were shown in the revised manuscript and Supplementary Information.

Comment (2) Page 2, line 26, and elsewhere: I think that nomenclature has to be corrected in some instances (see also below). Here, the term “*o*-biphenyl-substituted methylenefluorenes” should rather become “*o*-biphenyl-substituted methylenefluorenes”.

Response: Pages 2 and 13, “*o*-biphenyl” was corrected as “*o*-biphenyl”.

Comment (3) Page 3, lines 48-52: The sentence “In this study, the conjugated diene moiety in DFDPE was predicted to be more electron rich than the fluorenyl and phenyl moieties by the frontier molecular orbital calculation, which undergoes the double single electron oxidation preferentially to form a dication species, followed by the intramolecular Friedel-Craft reaction to afford the corresponding dispirocyclic DSFIIF.” contains several minor mistakes and has also wrong syntax. Rather, it should read: “In this study, the conjugated diene moiety in DFDPE was predicted to be more electron-rich than the fluorenyl and phenyl moieties by the frontier molecular orbital calculation, which suggests that it would easily undergo two-fold *single-electron* oxidation preferentially to form a dication species, followed by the intramolecular Friedel-Crafts reaction to afford the corresponding dispirocyclic DSFIIF.”

Response: Page 3, line-48-52; the sentence was corrected according to the reviewer’s comment; “In this study, the conjugated diene moiety in DFDPE was predicted to be more electron-rich than the fluorenyl and phenyl moieties by the frontier molecular orbital calculation, which suggests that it would easily undergo two-fold *single-electron* oxidation preferentially to form a dication species, followed by the intramolecular Friedel-Crafts reaction to afford the corresponding dispirocyclic DSFIIF.”

Comment (4) Page 3, line 59: The term “carrier-transporting materials” is incomplete, I assume that the authors mean “charge carrier-transporting materials”.

Response: Page 3, line 59, the term “carrier-transporting materials” was corrected as “charge carrier-transporting materials”.

Comment (5) Page 3, line 68: Also, I consider the phrase “prefunctionalisation of aromatic C-H bonds” inappropriate. It is not the “C-H bonds” but the “aromatic rings”, “aromatic units”, or aromatic nuclei” can be functionalised.

Response: page 3, line 68: the phrase “prefunctionalization of aromatic C-H bonds” was corrected as “prefunctionalization of aromatic rings”.

Comment (6) Page 4, Figure 1: Although a complete mechanistic interpretation will follow at the end of the manuscript (Figure 7), I find the schematic presentation of the reaction sequence not clear enough. (i) The label “1,2-aryl migration” in the third step is not sufficient; it should read “single electron oxidation and 1,2-aryl migration”. (ii) Also, the three steps (“single electron oxidation”, spirocyclization) and “single electron oxidation and 1,2-aryl migration”) should be complemented by adding “– e⁻”, “– H⁺”, and “– e⁻, – H⁺” near to

the reaction arrows.

Response: page 4, Figure 1b was modified; (i) the label “1,2-aryl migration” was corrected as “single electron oxidation and 1,2-aryl migration”. (ii) the three steps were modified by adding “– e⁻”, “– H⁺”, and “– e⁻, – H⁺” near to the reaction arrows.

Comment (7) Page 4, After “substrate **1a**”, it should read “(Table 1, see also Supplementary Information)”, instead of simply “(see Supplementary Information)”. The latter version would irritate the reader who immediately wants to know the identity of “substrate **1a**”.

Response: page 4, after “substrate **1a**”, “(Supplementary Information)” was modified to “Table 1, see also Supplementary Information”.

Comment (8) Page 5, line 88: “latter” (typo).

Response: page 5, “later” was corrected as “latter”.

Comment (9) Page 5, lines 96-97: “TFA” and “TfOH” should be named explicitly; thus, “trifluoroacetic acid (TFA)” and “trifluoromethylsulfonic acid (TfOH)” or, maybe, “triflic (TfOH)”. (It appears that “TfOH” does not appear anywhere later in the manuscript.)

Response: page 5, the full names of TFA and TfOH were provided; “trifluoroacetic acid (TFA)” and “trifluoromethanesulfonic acid (TfOH)”.

Comment (10) Page 5, line 108: I think that “has been investigated” should rather be “was investigated”.

Response: page 6: “has been investigated” was corrected as “was investigated”.

Comment (11) Page 6: There are several minor mistakes. (i) “Br and I at R¹” should be “Br and I as R¹”; (ii) “Similar electronic effects were ...” would be better (line 114); (iii) “Me and OMe at R²” should be “Me and OMe as R²”; (iv) “F and Cl at R²” should be “F and Cl as R²”; (v) “The BPMF **1i** with a perdeuterated *o*-phenyl group” would be better; (vi) benzothiophene-substituted” (hyphen).

Response: page 6: (i) “Br and I at R¹” was corrected as “Br and I as R¹”; (ii) “Similar electronic effect was” was corrected as “Similar electronic effects were”; (iii) “Me and OMe at R²” was corrected as “Me and OMe as R²”; (iv) “F and Cl at R²” was corrected as “F and Cl as R²”; (v) “deuterated” was corrected as “perdeuterated” in the new sentence related to the new substrate **1i-d₅**; (vi) hyphen was added between “benzothiophene” and “substituted”.

Comment (12) Page 7, line 131: Insert “the” before “synthesis”.

Response: page 8, “the” was added before “synthesis”.

Comment (13) Page 8, lines 135-136: “... composed of biphenyl-centered two methylene-fluorene moieties ...” is poor English; I suggest “... containing two biphenyl-centered methylene-fluorene moieties ...”.

Response: page 8, “composed of biphenyl-centered two methylene-fluorene moieties” was corrected as “containing two biphenyl-centered methylene-fluorene moieties”.

Comment (14) Page 8, lines 137-138: The sentence “It was noted ...” is too long; I suggest to delete the phrase “other than the corresponding product **1l**”.

Response: page 6: the phrase “other than the corresponding product **1l**,” was deleted after “It was noted that”.

Comment (15) Page 8, legend to Figure 3: "... biphenyl-centered two methylenefluorenes ..." is again poor English; I suggest to use rational chemical nomenclature, for example, "... biphenyl-centered bis-methylenefluorenes ...". The same holds true for part (b) of the legend.

Response: page 10, legend to Figure 3 was corrected by this comment and newly prepared products: (a) biphenyl-centered bis-methylenefluorenes, (b) *p*-terphenyl-centered bis-methylenefluorenes, (c) *m*-terphenyl-centered bis-methylenefluorenes, (d) 1,5-diphenylnaphthalene-centered bis-methylenefluorenes, (e) 1,1':5',1''-ternaphthalene-centered bis-methylenefluorenes.

Comment (16) Page 9, line 157: I do not understand the term "trisubstituted 2-vinyl-1,1'- biphenyl" for compound **1o**. Why not "triaryl-substituted ethene"? – Also, the term "2,2-diphenyl group" does not make sense. Maybe "diphenylmethylene unit" would be appropriate.

Response: page 10: "trisubstituted 2-vinyl-1,1'-biphenyl **1o**" was corrected as "triaryl-substituted ethene **1r**"; "2,2-diphenyl group" was corrected as "diphenylmethylene unit".

Comment (17) Page 9, line 160-161: The sentence "It was noted that other 2-vinyl-1,1'-biphenyls by replacement of 2,2-diphenyl in **1o** with other substituents, such as Ph and Me, Ph and H, dimethyl, and two hydrogens were examined to be totally inactive, demonstrating the importance of aryl groups for the implementation of the present tandem transformation." is again quite poor and should be improved.

Response: page 10: the commented sentence was modified as following; "Other biphenyl-substituted ethenes replacing the 2,2-diphenyl unit in **1r** with different substituents, such as Ph and Me, Ph and H, dimethyl, and two hydrogens were examined to be totally inactive, demonstrating the importance of triaryl groups for the implementation of the present tandem transformation."

Comment (18) Importantly (see also below), I consider the name "1,1'-biphenyl" for the biphenyl unit unnecessarily complicated. The biphenyl unit is necessarily constructed by a C-C bond between two carbons of each ring. Therefore, I recommend avoiding the term "1,1'-biphenyl" and replace it by "biphenyl" in the whole manuscript, including the Experiment section in the Supplementary Information.

Response: the term "1,1'-biphenyl" was replaced by "biphenyl" in the whole manuscript, including the Experiment section in the Supplementary Information.

Comment (19) Page 9, line 163: "Maybe "crucial for" ("the radical ion stabilization") would be better than "related to".

Response: page 10: "related to" was corrected as "crucial for".

Comment (20) Page 9, line 170: Delete "the" before "diluted".

Response: page 10: "the" was deleted before "diluted".

Comment (21) Page 10, "Mechanistic investigation" (I): I do not fully agree with the discussion concerning the role of the spirocyclic intermediate(s). – Of course, the finding that the spirocyclic hydrocarbon **3a** can be converted into the dibenzochrysenes **2a** supports – I would say, indirectly – the intermediacy of the related spirocyclic radical **B** shown in Figure 7. However, I think that the latter radical, being formed under the reaction conditions of the conversion **1a** → **2a**, does not undergo saturation (by H atom abstraction) to give **3a**; rather, **3a**

has to be oxidised and converted to either cation **C** (Figure 7) or, at least, to the radical cation **3a•+** in order to undergo the 1,2-aryl shift. Therefore, the relatively low yields of the oxidation reactions did proceed under the standard conditions, suggesting the formation of a spirocyclic radical intermediate in the present tandem reaction.” is a bit too simple and should be modified or completed according to my above arguments.

Response: according to this comment, the elucidation of Figure 6 was modified as following: “Although the yields of the desired product **2a** were relatively low, the oxidation reactions did proceed under the standard conditions, suggesting the formation of spirocyclic radical **B** and cation **C** intermediates (Figure 7) through the sequential single electron oxidation of **3a**. The formation of the cation species **C** may give rise to subsequent 1,2-aryl shift to produce the final product **2a**. It should be mentioned that, however, the present tandem reaction of **1a** to **2a** may not involve the formation of **3a** by the H atom abstraction.”.

Comment (22) Page 10, “Mechanistic investigation” (II): The information gained from the kinetic isotope effects is not discussed clearly. – (i) The KIE values of 0.9 and 0.98 are similar and close to unity, but they are also smaller than unity. What is the error limit? – (ii) Do those values mean that there is, actually, no kinetic isotope effect? In fact, this can be presumed for the mechanism suggested in Figure 7, with the C-C bond formation (or cleavage) steps are rate-determining. – (iii) However, if the KIEs are reliably smaller than unity, this would be a counter-intuitive result, which should be commented. – (iv) From the Experimental section in the Supplementary Information, it appears that the KIE values were determined from separated runs with **1a** and **1i**. In that case, I would assume rather substantial experimental uncertainties. However, if the reactions of **1a** and **1i** were done in the very same solution, then the reliability should be quite high. In any case, the experimental setup should be also stated more clearly.

Response: this is a pertinent comment for studying the isotope effect of the present reaction. In the original manuscript, we studied the isotope effect using the non-substituted BPMF **1a** and its perdeuterated **1i** as substrates. At that time, due to the superimposition of the ¹H NMR peaks of the corresponding two products **2a** and **2i**, we performed the isotope effect experiments in the separated reaction vessels under same conditions. As this reviewer commented, however, the isotope effect experiments carrying out under same reaction vessel should be more reliable for understanding the isotope effect. Therefore, in this revised experiments, we employed a new BPMF **1i** having two tert-butyl groups at the fluorene moiety and a methoxy group at the biphenyl moiety along with its perdeuterated **1i-d₅** as substrates for studying the isotope effect. Fortunately, the ¹H NMR yields of the corresponding products **2i** and **2i-d₄** were able to be calculated from the reaction mixture after performing the reaction in the same reaction vessel. The synthesis of **2i** and **2i-d₄** were shown in Figure 2 and the elucidation was added in page 6. The original perdeuterated **1i** was deleted in the revised manuscript.

Very similar KIEs of 1.21 and 1.2 were obtained using a 1:1 mixture of **1i** and **1i-d₅** in the same reaction vessel under the conditions A and B, respectively. The small isotope effect suggests that the present reaction involves a rapid aromatic deprotonation process. These isotope effects were elucidated in pages 11-12 according to the reviewer’s comment as following: “In addition, an intermolecular competing reactions between the protonated substrate **1i** and the perdeuterated substrate **1i-d₅** were studied in the same reaction vessel to

understand the kinetic isotope effects (KIEs) (Figure S2 in Supplementary Information). When a 1:1 mixture of **1i** and **1i-d₅** was heated at 40 °C for 1 h under conditions A and B, respectively, the corresponding products **2i** and **2i-d₄** were obtained with similar KIE values of 1.21 and 1.20. The small isotope effect suggests that the present reaction involves a rapid aromatic deprotonation process and in other words, the C-C bond formation or cleavage steps can be presumed to be the rate-determining.”

Comment (23) Page 12, lines 224-225: The sentence “In summary, we have disclosed a novel and efficient tandem synthetic method of various extended PAHs by the single electron oxidation of alkenes tethered with *o*-biphenyl and fluorine moieties.” is not clear and precise. – I would suggest writing “In summary, we have disclosed a novel and efficient tandem synthetic method for the construction of various extended PAHs by the single electron oxidation of various 1-*o*-biphenyl-2-fluorene-9-ylidenes.” or, alternatively, “In summary, we have disclosed a novel and efficient tandem synthetic method for the construction of various extended PAHs by the single electron oxidation of various (*o*-biphenyl)methylene-substituted fluorenes.”

Response: page 13: the commented sentence was modified as “In summary, we have disclosed a novel and efficient tandem synthetic method for the construction of various extended PAHs by the single electron oxidation of various (*o*-biphenyl)methylene-substituted fluorenes.”

Comment (24) Page 12, line 243, and throughout the Experimental section (SI): The term “white solid” should be replaced by “colourless solid”.

Response: the term “white solid” was corrected as “colourless solid” throughout the Experimental section (SI).

Supplemental Information

Comment (25) Page S2, 1st paragraph: Instead of “High-resolution mass spectra were obtained on a BRUKER APEXIII spectrometer and JEOL JMS-700 MStation operator.”, I recommend writing “Accurate mass data were obtained by high resolution mass spectrometry performed on a BRUKER APEXIII spectrometer and JEOL JMS-700 MStation operator.” (why the word “operator?”). In fact, the data obtained are accurate mass data; high-resolution is not an indispensable condition for obtaining accurate mass information; however, use of high resolution mass spectrometry is helpful if isobaric ions overlap.

Response: page 1: according this comment, the sentence in page S2 was modified as “Accurate mass data were obtained by high resolution mass spectrometry performed on a Bruker Daltonics solariX 9.4T FT-ICR-MS spectrometer or a Bruker Daltonics microflex instrument using Matrix Assisted Laser Desorption Ionization (MALDI) at the Research and Analytical Center for Giant Molecules, Graduate School of Science, Tohoku University.”

Comment (26) Page S2, 1st paragraph: The MALDI matrix used should be given also.

Response: The MALDI matrix used was given in the response (1) sentence.

Comment (27) Page S2, end of 2nd paragraph: The sentence “Other starting substrates and products were determined by ¹H, ¹³C NMR, and high-resolution mass.” has poor style. It should read “Other starting substrates and products were determined by ¹H and ¹³C NMR spectroscopy and high-resolution mass

spectrometry.”

Response: the sentence in page S2 was modified as “Other starting substrates and products were determined by ^1H and ^{13}C NMR spectroscopy and high-resolution mass spectrometry.”

Comment (28) Page S2, heading of 4th paragraph: Again, the naming of the biphenyl unit in the compound names should simply be “biphenyl” rather than “1,1’-biphenyl” – throughout the whole SI.

Response: “1,1’-biphenyl” was replaced by “biphenyl” throughout the whole SI.

Comment (29) Page S2, line 1 of text: To be strict, I wonder whether cupric chloride (CuCl) is soluble in *o*-xylene. However, I assume that it is soluble in a mixture of *o*-xylene and TFA.

Response: Yes, CuCl in a mixture of *o*-xylene and TFA. with TFA is soluble after heating.

Comment (30) Page S2, line 7 of text: Once again, please note here and below the “white solid” should be changed into “colourless solid”.

Response: the “white solid” was changed into “colourless solid” throughout the whole SI.

Comment (31) Page S3, scheme to 2nd paragraph: I recommend changing the poor style of the aldehyde group (“ =O ”) into the better one (“ -CHO ”), as it appears in the next paragraph, for example.

Response: page S3: the aldehyde group “ =O ” was rewritten as “ -CHO ”.

Comment (32) Page S5: From the presentation of the experiments, it appears indeed that two sets of two separate runs were performed. If this is really true, then the experimental uncertainties of the KIE values obtained should be rather substantial.

Response: page S6, Figure S2: as mentioned this reviewer, the KIE experiments were performed in the same reaction vessel by using a 1:1 mixture of **1i** and **1i-d₅**. The details were responded in the comment (22).

Comment (33) Page S7, and throughout: To be strict again, I recommend making clear whether the calculated accurate mass values rely to the cationic species, e.g. $\text{C}_{26}\text{H}_{18}^+$ or, rather, $\text{C}_{26}\text{H}_{18}^{+\bullet}$ in the present case, or to the neutral species (which would be less appropriate).

Response: the accurate mass values were corrected as the cationic species throughout the whole SI, e.g. $\text{C}_{26}\text{H}_{18}^+$.

Comment (34) Page S7, and throughout: The experimentally obtained accurate mass data appear to be excellent throughout. In such cases, I would prefer to see the next decimal (fifth) to have information about the error limit. This is only a recommendation.

Response: The fifth decimal values of experimentally obtained and calculated accurate mass data of all compounds were shown throughout the whole SI.

This is a long list of to-be-improved items, which is given to improve what is going to be an excellent paper. I think the paper does deserve this effort.

REVIEWERS' COMMENTS:

Reviewer #1 (Remarks to the Author):

I am happy to see this revised manuscript, where the authors have adequately addressed my previous concerns. In particular the added syntheses of new extended and intriguing PAHs highlight the strength and potential of this method. I would consider that the proposed reaction mechanism is not yet unambiguous, but the experiments and discussion by the authors are reasonable and of high quality, which should be sufficient for this work. I would recommend the publication of this manuscript in Nature Communications after one minor correction: “Bandgap” and “optical bandgap” used for PAHs should be corrected to “HOMO–LUMO gap” and “optical (energy) gap”, respectively.

Reviewer #1 (Remarks to the Author)

Comment: I am happy to see this revised manuscript, where the authors have adequately addressed my previous concerns. In particular the added syntheses of new extended and intriguing PAHs highlight the strength and potential of this method. I would consider that the proposed reaction mechanism is not yet unambiguous, but the experiments and discussion by the authors are reasonable and of high quality, which should be sufficient for this work. I would recommend the publication of this manuscript in Nature Communications after one minor correction: “Bandgap” and “optical bandgap” used for PAHs should be corrected to “HOMO–LUMO gap” and “optical (energy) gap”, respectively.

Response: In page 10, line 201; page 11, lines 209 and 211, “optical bandgap” was corrected to “optical energy gap”. In page 11, line 203, “bandgaps” was corrected to “HOMO-LUMO gaps”.

All the comments of the reviewers were meaningful for improving the original manuscript; I thank them for spending their valuable time to review our manuscript carefully and in constructive manner. I hope that the revised manuscript will meet the usual high standard set to Nature Communications.